# Role of Cardiac Macrophages on Cardiac Inflammation, Fibrosis and Tissue Repair

**DOI:** 10.3390/cells10010051

**Published:** 2020-12-31

**Authors:** William P. Lafuse, Daniel J. Wozniak, Murugesan V. S. Rajaram

**Affiliations:** 1Department of Microbial Infection and Immunity, College of Medicine, Ohio State University, Columbus, OH 43210, USA; William.Lafuse@osumc.edu (W.P.L.); Daniel.Wozniak@osumc.edu (D.J.W.); 2Department of Microbiology, Ohio State University, Columbus, OH 43210, USA

**Keywords:** cardiac macrophages, fibrosis, tissue repair, cardiac inflammation

## Abstract

The immune system plays a pivotal role in the initiation, development and resolution of inflammation following insult or damage to organs. The heart is a vital organ which supplies nutrients and oxygen to all parts of the body. Heart failure (HF) has been conventionally described as a disease associated with cardiac tissue damage caused by systemic inflammation, arrhythmia and conduction defects. Cardiac inflammation and subsequent tissue damage is orchestrated by the infiltration and activation of various immune cells including neutrophils, monocytes, macrophages, eosinophils, mast cells, natural killer cells, and T and B cells into the myocardium. After tissue injury, monocytes and tissue-resident macrophages undergo marked phenotypic and functional changes, and function as key regulators of tissue repair, regeneration and fibrosis. Disturbance in resident macrophage functions such as uncontrolled production of inflammatory cytokines, growth factors and inefficient generation of an anti-inflammatory response or unsuccessful communication between macrophages and epithelial and endothelial cells and fibroblasts can lead to aberrant repair, persistent injury, and HF. Therefore, in this review, we discuss the role of cardiac macrophages on cardiac inflammation, tissue repair, regeneration and fibrosis.

## 1. Introduction

Macrophages are the central regulator of immune systems, able to activate and proliferate lymphocytes to generate innate and adaptive immune response [1]. The cardiovascular system consists of the heart, blood vessels, and blood, which transports nutrients, oxygen, and hormones to cells throughout the body and removes metabolic wastes. Heart failure (HF) is the clinical indication of various forms of cardiovascular diseases (CVDs) that impact the function of the heart. CVD includes atherosclerosis, ischemic heart disease, cerebrovascular disease, hemorrhagic stroke, hypertensive heart disease, cardiomyopathy, myocarditis, atrial fibrillation, aortic aneurysm, peripheral vascular disease, and endocarditis [2]. According to 2016 mortality data, the World Health Organization reports that CVD caused more than 17.9 million deaths worldwide, which is more than cancer and chronic lower respiratory disease combined [3]. Generally, regeneration of adult heart tissue after injury is poor. Therefore, understanding how cardiac tissue is injured and how cardiac tissue regenerates or resolves the damage is of prime importance to universal health.

Like other organs, the heart is composed of a heterogeneous population of cells including cardiomyocytes, fibroblasts, pericytes, smooth muscle cells, endothelial cells, and many types of immune cells (Figure 1) [4,5]. The immune system provides a protective inflammatory response necessary for host defense from infections and also plays a critical role in resolving local damage caused by sterile inflammation in the heart. Macrophages are the central regulator of immune systems [1], and the primary immune cells that reside in the heart tissue during steady state [5,6]. A small number of monocytes and a sparse population of dendritic cells are also present. Mast cells, regulatory T and B cells are also found in resting cardiac tissue, which helps to initiate the early immune response [5]. Generally, neutrophils are not present in healthy heart tissue [5], but they are recruited upon tissue injury or infection. These cardiac immune cells are activated by either sterile inflammation or microbial infections which leads to the production of inflammatory cytokines and recruitment of diverse leukocyte populations into inflamed heart tissue. Intracellular signaling and cross talk between embryonic tissue-resident cardiac macrophages (CMs) and non-embryonic CMs are critical in the generation, propagation and development of cardiac inflammation, tissue remodeling and repair [5]. Herein, we review the current knowledge of the contribution of tissue-resident cardiac macrophages on cardiac inflammation, fibrosis, and tissue repair.

## 2. Cardiac Immune Cells

The heart is immunologically active even during steady state and contains all major leukocyte populations, either residing in the heart or waiting to infiltrate into the heart. There are approximately 10^3^ leukocytes/mg of tissue in an adult mouse heart [7] (Table 1).

The majority of leukocytes present in the resting heart are F4/80^+^CD11b^+^ macrophages [5], with small populations of dendritic cells, B cells, regulatory T (Tregs) cells, and innate lymphoid cells [9,11,12,13]. Upon ischemic or non-ischemic cardiac injury, necrotic cell death leads to the activation of tissue-resident immune and non-immune cells. The danger-associated molecular patterns (DAMPs) released from necrotic cells are the key trigger for immune cell activation, which results in the release of pro-inflammatory cytokines and chemokines that are responsible for the recruitment of inflammatory leukocytes from the blood. Following the initial inflammatory phase, the expansion of neutrophil and macrophage populations facilitates phagocytosis and clearance of the dead cells, and release of cytokines and growth factors. This leads to the initiation of the healing process via the activation of myofibroblast proliferation and neovascularization of the injured myocardium [14].

Monocytes are another vital part of the innate immune system during both the initial insult and the chronic phase of cardiac injury. In steady-state conditions, the heart contains few monocytes. In humans, there are three distinct monocytic subsets which have been identified based on the expression of CD14 and CD16, namely classical (CD14^++^ CD16^−^); intermediate (CD14^++^ CD16^+^) and non- classical (CD14^+^ CD16^++^) monocytes [15]. In contrast, murine monocytes have been classified into two subsets based on their expression of Ly6C; classical Ly6C^high^, CCR2^high^ and chemokine (C-X3-C motif) receptor-1 (CX3CR1)^low^ and non-classical Ly6C^low^ CCR2^high^CX3CR1^high^ monocytes. The classical monocytes are recruited and accumulate at inflammatory sites, including during acute myocardial infarction (MI). Conversely, non-classical monocytes patrol the endothelium to maintain homeostasis [16]. HF in humans has also been associated with peripheral monocytosis [17] and exaggerated inflammation with distinct monocyte profiles [16,18]. In response to cardiac stress, Ly6C^high^ monocytes are rapidly recruited to the mouse heart, either following ischemic injury or hypertensive stress; in contrast Ly6C^low^ monocytes do not get recruited into the myocardium during the early inflammatory stage but appear in the heart during the later reparative stage [6,19,20,21]. The antigen presenting dendritic cells (DCs) are dispersed throughout the heart [22], which links the innate immunity to adaptive immunity. At steady state, DCs function in maintaining\peripheral tolerance [23,24] by presenting self-antigens to Tregs [25]. The heart contains two conventional DC subsets CD103^+^ CD11b^−^ (cDC1) and CD103^−^CD11b^+^ (cDC2) [12,26,27]. Both cDC subsets are of hematopoietic origin and dependent on Flt3L for development [12] and these DCs differ from CMs by lack of expression of CD64 and myeloid-epithelial-reproductive tyrosine kinase (MERTK) [12]. Both cardiac DCs express the chemokine receptor CCR2 and *Ccr2*-deficient mice result in reduced numbers of cDCs in the myocardium of the heart, indicating that CCR2 is a key regulator of the migration of pre-cDCs into the heart [12]. Recently, it has been demonstrated that cardiac-resident MHCII^+^ cells process and present myosin heavy chain alpha-derived peptides under steady-state heart function and have the capability to prime T cells ex vivo [28,29]. In addition, previous studies demonstrated that CD4 T cell-deficient mice, and the transgenic mice that express a single T cell receptor gene to an irrelevant peptide, show impaired wound healing and increased expansion of monocyte populations after ischemic injury, suggesting that CD4 T cells have protective functions during cardiac injury [30]. Interestingly, these cardiac CD4^+^ T cells are activated by presentation of self-antigens via MHCII expressing cells in the heart. This leads to immunosuppressive responses in the myocardium, suggesting that most CD4^+^ T cells in the heart are regulatory T cells (Tregs) that promote myocardial recovery through an IL-10-dependent pathway [31,32]. Additionally, the recruitment and proliferation of regulatory T cells (Foxp3^+^CD4^+^) in the heart improves myocardial wound healing after MI by modulating monocyte/macrophage differentiation [10]. Another study by Yang et al. reports that the Rag-1 knock out mice (do not have mature T and B cells) develop smaller infarcts than control C57BL/6 mice, and transfer CD4^+^T cells into the RAG-deficient mice increased the infract size and healing [33]. Interestingly, recent studies revealed that transverse aortic constriction can induce T cell responses, and the recruitment of CD4^+^ T cells contributes to the pathogenesis of HF [34,35].

## 3. Cardiac-Resident Macrophages

Macrophages are specialized cells of the immune system that recognize, phagocytose and destroy apoptotic ells and pathogens and generate innate immune response. The major challenge in the field of cardiac macrophages is the discrepancy among the macrophage phenotypes and their markers identified and reported by various groups. In this review, we consolidated the CM phenotypes and common markers reported in the literature for the CM classification and function. These macrophages are dispersed throughout the heart as spindle-like cells, acting as sentinel macrophages for pathogens that may enter the myocardium. In the past few decades, it has been universally accepted that bone marrow-derived hematopoietic stems cells differentiate into circulating monocytes which enter into various tissues and differentiate into tissue-resident macrophages [36,37]. However, in recent years, the understanding of the origin of tissue-resident macrophages has been drastically revised with the recognition that resident macrophage populations are established during embryo development and maintain their numbers through self-renewal properties, rather than through infiltration of blood monocytes [6,19,38,39,40,41]. The heart contains macrophage subsets with distinct functions and origins. Using flow cytometry, linage tracing, and parabiosis studies, subsets of chemokine (C-C motif) receptor-2 (CCR2)^−^ and CCR2^+^ CMs have been defined [19,41,42,43,44]. (Table 2) CCR2^−^ CMs are derived from the embryonic yolk sac and maintained without monocyte recruitment.

In contrast, CCR2^+^ cardiac macrophages are maintained by monocyte recruitment. These studies also show that the CCR2^−^and CCRR2^+^ CM differ in function, with the CCR2^−^CMs being involved in maintenance of homeostasis and the CCR2^+^ being pro-inflammatory. Furthermore, studies by Epelman et al. [6] characterized the CMs into three subsets defined by the difference in expression levels of MHC class II, CCR2 and CD11c (Table 2). The predominant cardiac macrophages were two CCR2^−^ populations (MHCII^high^ CD11c^low^ and MHCII^low^ CD11c^low^). These two subsets were primarily derived from yolk sac progenitors and renewed through in situ proliferation. The third cardiac macrophage subset was CCR2^+^, MHCII^high^ and CD11c^high^ were derived and slowly replenished from circulating monocytes and these CMs were distinguished from dendritic cells by expression of macrophage markers CD64 and MerTK [47] (Figure 2).

The transcript profile analysis of the two CCR2^−^ macrophage populations (MHCII^high^ and MHCII^low^) and the CCR2^+^ MHCII^high^ populations indicated overlapping and non-overlapping functions. The CCR2^+^ macrophages were enriched for inflammatory genes and pathways, including genes involved in the NLPR3 inflammasome and IL-1β production. Consequently, this suggests that these macrophages are inflammatory in nature. The MHCII^high^ subsets were enriched for genes involved in antigen presentation to T cells, suggesting involvement in immune surveillance. The uptake of apoptotic/necrotic cells by CCR2^−^ MHCII^low^ suggests these macrophages function in homeostasis during steady state by removing dead cells without inducing an immune response [48]. Moreover, single-Cell RNA sequencing by Dick et al. also identified four different cardiac macrophage clusters at steady state, each with unique functions [46] (Table 2). One macrophage cluster identified as CCR2^−^ TIMD4^+^ LYVE1^+^ MHCII^low^ was maintained independent of blood monocytes and corresponds to the CCR2^−^ MHCII^low^ subset [6]. This cluster of macrophages expressed the genes involved in homeostasis and regeneration, including phosphatidylserine receptor *Tmid4*, *Lye1* and growth factor *Ifg1,* which is also involved in efferocytosis of dead cells [49], vascular homeostasis [50] and angiogenesis [43], respectively. The second cardiac macrophage clusters were CCR2^−^MHC^high^ TIMD4^−^ LYVE1^−^ and were partially replaced by blood monocytes in long-term parabiosis. One of these clusters expressed high levels of genes involved in antigen presentation and thus correlates with the CCR2^−^ MHC^high^ macrophage subset [6]. The third cluster, termed the *Isg* cluster, was enriched in genes stimulated by interferon such as *Irf7, Isg20,* and *Ifit1.* The fourth cluster contained CCR2^+^ macrophages, which were fully replaced by blood monocytes in long-term parabiosis. Studies of the human heart have described analogous resident CCR2^−^and CCR2^+^ cardiac macrophages [44,46] as functionally distinct macrophage subsets that exist in the human heart at steady state. Cardiac macrophages are also abundant in the atrioventricular node (AV), which provide an electrical connection between the atria and ventricles while also facilitating an electrical conduction by coupling to beating CMs via connexin-43-containg gap junctions [46].

## 4. Macrophage Role in Cardiac Inflammation and Cardiac Dysfunction

Myocarditis is defined as an inflammation of the myocardium in the presence of necrosis or degeneration of cardiomyocytes and other cell types. Myocarditis accounts for about one in nine cases of HF and remains one of the top reasons for heart transplantation worldwide, since the lack of specific treatments [51]. The most common causes of myocarditis are infections of various pathogens including parasite *Trypanosoma cruzi* in South America, or viruses, such as enterovirus, parvovirus B19, adenovirus and hepatitis C virus, which are the major causes in North America [52,53,54]. Recently, we and others have shown that bacterial infection also induces myocardial inflammation and cardiac dysfunction [55,56,57,58,59,60]. In addition, non-infectious agents, such as allergic reactions to drugs and chemicals, can induce cardiomyocyte apoptosis and the release of DAMPs, which activate the innate immune response leading to sterile inflammation and myocarditis [53].

Viral infections are the most common cause of myocarditis, with periodicity to this prevalence in the population and possibly related to herd immunity. The pathophysiology of viral myocarditis involves both a direct virus-mediated response and indirect immune-mediated injury of the cardiac tissue and subsequent dysfunction. The innate immune response is essential for the elimination of virus and the restoration of the normal tissue homeostasis. Nonetheless, activation of the immune system also leads to myocardial inflammation, necrosis, release of DAMPs, and ventricular dysfunction, which occurs directly through the killing of virus-infected cardiac cells, and also through an increased production of inflammatory cytokines which subsequently affect CM function [54].

The process of virus-induced myocarditis includes three stages, an acute viremia stage, followed by sub-acute infiltrating stage, and a chronic stage [61]. In the acute stage, a virus enters into cardiomyocytes or other cells (endothelial cells, fibroblasts and immune cells) through specific surface receptors, i.e., the coxsackievirus binds to coxsackievirus and adenovirus receptor (CAR) and the co-receptor, the decay accelerating factor (DAF). Viral entry activates the inflammatory signaling cascade and produces various pro-inflammatory cytokines (IL-1, TNF, IFN-α, IFN-β) and chemokines, resulting in recruitment of immune cells into the tissue and an aggravation of the inflammatory process [61]. During the infiltration stage, the migrated immune cells activate B cells to produce neutralizing antibodies and promote viral clearance [62]. Finally, in the chronic stage, once the virus is cleared completely, apoptotic cell debris is removed by macrophages and the tissue restoration process is initiated and leads to cardiac fibrosis (discussed below). However, in some cases the viral RNA and proteins can persist in the heart and cause chronic inflammation, which triggers extensive fibrosis and ventricular dysfunction and can eventually develop into dilated cardiomyopathy [61]. The infiltrating activated immune cells into the myocardium increase the expression of matrix metalloproteinase (MMP)-9, which in turn exacerbates the immune cell recruitment and refuels the inflammatory process [63]. Several human studies have demonstrated an increase in the activity and expression of MMP-2 and MMP-9 in virus-induced dilated cardiomyopathy patients [64]. Since MMPs play prominent role in the inflammation and remodeling processes of the myocardium, great interest arises concerning the development of pharmacological inhibitors of MMPs that might be useful for the treatment of viral myocarditis and dilated cardiomyopathy.

The coronavirus infectious disease 2019 (COVID19) is caused by the severe acute respiratory syndrome coronavirus 2 (SARS-CoV2), a single-stranded RNA virus of the Coronaviridae family. Initially described as a respiratory syndrome [65], clinical studies indicate involvement of multiple organ systems in COVID19, including the heart [65,66,67]. COVID19 patients frequently show significant myocardial damage as evidenced by elevated troponin-I levels, a biomarker of cardiac damage [68,69,70]. COVID19 patients may also present other cardiovascular complications, such as myocarditis, acute myocardial infarction, heart failure, arrhythmias, and venous thromboembolic events [69,71,72], and almost half of mildly ill COVID19 patients have abnormalities in heart function [73,74,75,76]. Together these clinical studies suggest that SARS-CoV2 can directly infect cells in the heart. Indeed, in situ hybridization of cardiac tissue from autopsy cases detected SARS-CoV-2 virus in heart interstitial cells [76]. However, this study did not identify the interstitial cells that were infected with the virus. Interestingly, this study found that the presence of virus in the heart was not associated with increased infiltration of mononuclear cells into the myocardium. Post-mortem studies of a child with COVID19-related multisystem inflammatory syndrome found myocarditis characterized by inflammatory cell infiltration associated with foci of cardiomyocyte necrosis. Electron microscopy detected viral particles present in several cell types including cardiomyocytes, endothelial cells, macrophages, neutrophils, and fibroblasts. The authors of this study proposed that infection of cardiomyocytes leads to cardiomyocyte necrosis and a local inflammation response, resulting in spread of the virus to other cell types [77].

Other studies demonstrated that SARS-CoV2 can enter and replicate within human-induced pluripotent stem cell (hiPSC)-derived cardiomyocytes and induce cytotoxic effects that abolish cardiomyocyte beating [78,79,80]. SARS-CoV-2 enters cells by binding of the viral Spike protein (S) to the SARS-CoV2 receptor ACE2 and S priming by the serine protease TMPRSS2 [81]. Studies by Perez-Bermejo et al. [80] examined the expression of ACE2 and TMPRSS2 in hiPSC-derived cardiomyocytes. ACE2 was expressed by the hiPSC-derived cardiomyocytes but not TMPRSS2. They showed that SARS-CoV2 employs the ACE2 receptor to enter cardiomyocytes and utilizes the cathepsin-L endolysomal pathway to prime the S protein. Infection of human-iPSC-derived cardiomyocytes with SARS-CoV2 caused sarcomere fragmentation and transcriptomic changes in the cardiomyocytes. Although these studies indicate that SARS-CoV2 can enter and replicate within cardiomyocytes, the infectivity of resident cardiac macrophages is largely unknown. This is important in that resident cardiac macrophages are in close proximity to the cardiomyocytes and have cell–cell contact [82,83]. The influence of SARS-CoV2-induced hypoxia on cardiomyocytes, the systemic cytokine storm induced by the virus, and psychological stress likely also has significant impact on cardiac damage in COVID19 patients.

Chagas disease, caused by the protozoan *Trypanosoma cruzi*, is very prevalent in South and Central America [84]. Chagas disease consists of an acute phase and chronic phase. The acute phase of infection causes inflammation at the site of infection, lymphadenopathy, and hepatosplenomegaly. In the acute phase of the infection, the parasite load dictates the magnitude of the inflammatory response and cardiac tissue damage by the host immune response and the parasite itself [85,86,87]. During the severe acute phase, patients develop acute myocarditis, pericardial effusion and meningoencephalitis, which increases the mortality rate [88]. The acute phase usually resolves spontaneously over time. However, if the infection is untreated a chronic phase develops. More than 30–40% of chronically infected patients develop cardiac dysfunction, which mainly affects the conduction system and myocardium [89,90,91]. Severe manifestation leads to sinus node dysfunction, bradycardia, high-degree atrioventricular block, ventricular tachycardia and dilated cardiomyopathy with congestive HF [92]. Several experimental models of the *T. cruzi* infection suggest that CD4^+^ and CD8^+^ T cells-mediated Th-1 cytokine (IFN-γ, TNF-α, IL-12 and IL-18) response is important to control the parasites, while an uncontrolled production of these cytokines promotes myocarditis by inducing apoptosis of the cardiomyocytes and other immune cells [93,94,95]. Notably, *T. cruzi* infection in the heart enhances the production of IFN-γ, which increases T cell migration and the establishment of myocarditis by promoting the expression of chemokines (CCL5, CCl2, CXCL10 and CXCL9) and chemokine receptors, intercellular adhesion molecules and the vascular cell adhesion molecule [96,97]. In addition, chemokines play a critical role in elimination of parasites through the induction of nitric oxide production by *T. cruzi*-infected cardiomyocytes and macrophages, and thus contributing to the pathogenesis of Chagas cardiomyopathy [98].

Bacterial myocarditis is an uncommon cause of infectious myocarditis, mainly due to overwhelming sepsis or as part of a specific bacterial syndrome. The most common bacterial cause of myocarditis is *Staphylococcus aureus*, which causes myocardial abscesses and bacterial endocarditis [99,100]. In recent years, infection of Gram-positive bacteria enterococci species are responsible for 8–17% of infective endocarditis cases [101,102,103]. The infective endocarditis caused by enterococci spp. is becoming more prevalent in elderly patients with degenerative heart valve disease, prosthetic heart valves, and a higher incidence of enterococci bloodstream infections originating from the gastrointestinal or urogenital tracts [102,104,105,106]. The other bacteria that cause acute myocardial infarction and severe HF are *Streptococcus pneumoniae*. Brown et al. [55], demonstrated that *S. pneumoniae* disseminates into the bloodstream and translocate into the myocardium, where the bacteria cause microlesions. These microlesions are found in both left and right ventricles and adjacent to blood vessels and leads to the development of abnormal electrophysiology. Bacterial translocation is facilitated by pneumococcal adhesion molecules CbpA binding to the host ligands, laminin receptor and platelet activating factor receptor [55,56]. Additionally, this study revealed that the bacterial toxin pneumolysin binds to eukaryotic cell membrane, oligomerizes, and forms lytic pores. The formation of lytic pores can lead to lysis of the target cell and have a disruptive effect on cell signaling, as well as altering the entry of Ca^2^+ or causing loss of small molecules such as ATP [55,107]. The cardiomyocyte contractile function is dependent on calcium release; the release of pneumolysin dysregulates the calcium influx into the cardiomyocytes and thereby induces contractile dysfunction. On the other hand, pneumolysin can directly act on cardiomyocytes and other immune cells in the myocardium to induce apoptosis, which does not depend on IL-1β-mediated pyroptosis [55]. The necrotic cell death induced by pneumolysin typically elicits a strong inflammatory response due to the release of DAMPs, which recruit activated immune cells leading to the induction of cardiac fibrosis.

Other bacterial infections that lead to myocardial inflammation and cardiac dysfunction are *Corynebacterium diphtheriae*, *Francisella tularensis* sp *novicida*, and *Mycobacterium avium* species. Diphtheria toxin produced by *C. diphtheriae* inhibits elongation factor-2 activity in protein synthesis and causes DNA fragmentation and cytolysis, resulting in both local and systemic manifestations [108]. This leads to myocarditis, neuritis and focal necrosis in various organs. Diphtheria toxin-mediated myocarditis is mainly due to damage to muscle fibers, which progresses to myolysis with fibrosis and resulting in permanent cardiac damage, without the involvement of immune cells [109,110]. Our previous studies demonstrated that *Ft. novicida* infection disseminates to heart tissue, causing myocardial lesions via inducing cardiac inflammation, cardiomyocyte apoptosis, and immune cell recruitment into the heat. Due to increased myocardial cell apoptosis and clearance of dead cells, cardiac fibrosis is induced and causes severe damage to cardiac electrical conduction [60]. Furthermore, we demonstrated that the *M. avium* infection in aged mice enhances the recruitment of CD45^+^ leukocytes into the heart and increased expression of inflammatory cytokines, which results in the induction of cardiac fibrosis and cardiac hypertrophy [59]. In addition, many bacterial, fungal and viral infections cause inflammation in the endocardium of the heart [111]. Several studies demonstrated that both infectious pathogens and non-infectious agents can cause acute pericarditis, an inflammation in the pericardium of the heart [112].

There are multiple causes for myocardial infarction that lead to heart failure, including the extensive area of myocardium damage and excessive inflammatory response that induces reactive fibrosis and remodeling of the ventricular wall outside the infarction [113]. This reactive fibrosis is associated with cardiomyocyte hypertrophy to compensate for increased workload by an increase in size. Excessive secretion of pro-fibrotic factors during myocardial infarction can also lead to leakage into surrounding areas of the myocardium resulting in proliferation of local fibroblasts and collagen deposition. Incomplete and delayed resolution of the inflammatory phase of myocardial infarction may also lead to late phase remodeling and heart failure [114].

## 5. Activation of Cardiac Macrophages in Diseased Heart

Macrophages comprise the innate and adaptive immune system with a major role in immune system defense, inflammation and tissue restoration. Cardiac tissue contains large numbers of resident macrophages, further increased by infiltration of circulating monocytes during injury [6]. These resident macrophages are activated by the recognition of pathogen/damage-associated molecular patterns (PAMPs/DAMPs) by a number of pattern recognition receptors (PRRs) [115]. For example, lipopolysaccharides (LPS) from Gram-negative bacteria, lipoteichoic acid from Gram-positive bacteria, zymosan from fungi, lipoarabinomannan from Mycobacteria, bacterial flagellin and toxins, viral and bacterial CpG DNA and single-stranded RNA from viruses are sensed by various PRRs [116]. The PRRs are divided into two groups based on their subcellular localization: toll-like receptors (TLRs) and C-type lectin receptors are present on plasma membranes or endosomes and the second class of PRRs, including retinoic acid inducible gene –I- like receptors (RLRs), nucleotide binding and oligomerization domain (NOD)-like receptors and absent-in-melanoma-2 (AIM2) receptors, are localized in intracellular compartments [117,118]. In the heart, both PAMPs and DAMPs are detected by membrane PRRs and mediate the signaling cascades that activate nuclear factor-kB (NF-κB), activation protein-1 (AP1), and interferon regulatory factors (IRFs) transcription factors, that in turn enhance the expression of genes that encode pro-inflammatory cytokines and interferon in the heart [119]. The activation of intracellular PRRs in the heart leads to distinct pro-inflammatory signaling complexes called inflammasomes, which convert pro-caspase-1 into the catalytically active protease that is responsible for the production of interleukin-1β (IL-1β) and IL-18, subsequently triggering cardiac inflammation [120].

TLRs 1–10 are expressed in the human heart and particularly TLR2 and TLR4 are expressed abundantly [121], though the expression of TLRs have not been identified in human myocytes. Frantz et al., demonstrated that TLRs (2, 3, 4 and 6) are expressed in the myocytes of neonatal rats [122]. Although regulation of TLR expression in the heart has not been studied fully, TLR4 expression seems to be upregulated in the failing human heart and on circulating monocytes at the time of myocardial infarction [123,124]. Studies also show that loss of TLR2 in hematopoietic cells provide protection during ischemic injury [125].

The heart can be injured through a variety of pathophysiological processes, which can be grouped broadly into ischemic and non-ischemic etiologies—acute ischemic injury is the best studied model of cardiac injury and repair. Following injury, necrotic cell death leads to the activation of leukocyte populations in the cardiac microenvironment, which initiates an inflammatory response characterized by the production of pro-inflammatory cytokines and chemokines by resident immune and non-immune cells that are responsible for the recruitment of leukocytes into the area of injury. DAMPs which include adenosine triphosphate (ATP), several members of heat shock proteins (HSPs), S100A8/9, high mobility group protein B1(HMGB1), hyaluronic acid, fibrinogen, fibronectin, β-defensin, neutrophil elastase, and histones released from dying cells activate the resident macrophages, which leads to the production of inflammatory cytokines and causes sterile inflammation in the heart [115,126]. The sterile inflammation caused by non-infectious DAMPs can cause myocarditis in susceptible individuals. The sterile inflammation can be induced following cardiac injury stimuli such as myocardial infarction, cardiac surgery, allergic reactions to drugs or chemicals or excessive stress, which can be reproduced in the laboratory by exposure to myosin [127]. In addition to PAMPs and DAMPs, the cytokines released from inflammatory cells in the myocardium activate the resident macrophages, and play a critical role in macrophage polarization. For example, clearance of dead cells from the infarcted heart enhances the production of IL-10 and TGF-β, which in turn promote the anti-inflammatory M2 phenotype of CMs. In addition, secreted pro-inflammatory cytokines act on macrophages and promote the inflammatory phenotype [128].

## 6. Role of Cardiac-Resident Macrophages in Tissue Repair during Cardiac Injury

Myocardial infarction results from blockage of coronary arteries that supply oxygen to the heart. Ischemic injury occurs in the region of the heart that is normally oxygenated by the blocked artery. Compared to adult mice, neonatal mice have striking ability to repair cardiac injury by regenerating the myocardium [129,130]. Studies by Lavine et al. [42] showed that the neonatal mice heart contains one macrophage population (MHCII^low^ CCR2^−^) and one monocyte (MHCII^low^ CCR2^+^) subset, in contrast the adult heart contains two different resident CMs subsets (MHCII^low^ CCR^−^ and MHCII ^high^ CCR2^−^). Upon cardiac injury in neonatal mice, an embryonic-derived resident CCR2^−^ MHCII^low^ macrophage population expanded and did not recruit CCR2^+^ monocytes. In contrast, cardiac injury in the adult heart recruited monocytes and MHC-II^high^ CCR2^+^ monocyte-derived macrophages. CCR2^−^ macrophages isolated from neonatal hearts were reparative with the ability to stimulate cardiomyocyte proliferation and angiogenesis, and minimal ability to induce an inflammatory cytokine response. Whereas CCR2^+^ macrophages isolated from adult hearts produced a strong inflammatory cytokine response to LPS and were unable to stimulate cardiomyocyte proliferation and angiogenesis. Thus, these studies indicate that resident CCR2^−^ MHCII^low^ macrophages are key mediators of cardiac repair in the neonatal heart.

At steady state, the adult mouse heart also contains embryonic-derived resident CCR2^−^ MHCII^low^ macrophage with reparative properties similar to the neonatal macrophages [42]. However, CCR2^−^ resident macrophages in adult mice are rapidly lost within infracted myocardium, being reduced by 60% at day 2 after infarction [131]. The loss of resident CMs is largely due to cell death with a significant increase in dead macrophages being observed within 2 h of infarction [132]. The resident CCR2^−^ macrophages are replaced by an influx of inflammatory CCR2^+^ monocytes and monocyte-derived macrophages into the infarcted tissue.

Necrotic cardiomyocytes and macrophages release DAMPs that trigger a rapid inflammatory response in the injured myocardium. The interaction of DAMPS with Toll-like receptors activates inflammatory signaling pathways leading to the production of various chemokines: CCL chemokines (CCL-2, CCL5 and CCL-7) to attract leukocytes and mast cells, and CXC chemokines (CXCL1, 2, and 8) to attract neutrophils [133,134]. A cascade of pro-inflammatory cytokines such as TNF-α, IL-1β, IL-6 and IL-18 further amplify the inflammatory response by activating the resident and invading leukocytes [115] to clear the cellular debris and dead cells [134,135]. CMs can also detect extracellular DNA released from dying cells by a cGAS-STING/IRF3 pathway that results in a Type I interferon response [136]. Release of type I interferons can then induce interferon-stimulated gene expression by autocrine and paracrine activation which can amplify the inflammatory response in the infracted heart. Granulocyte-macrophage colony-stimulating factor (GM-CSF) is also produced by cardiac fibroblasts after myocardial infarction and can act locally to promote recruitment of monocytes and distally to induce production of Ly6C^high^ monocytes in the bone marrow [137].

CCL2 is essential for migration of inflammatory CCR2^+^ monocytes into the infracted tissue. CCL2^−/−^mice have reduced monocyte infiltration, lower interstitial fibrosis, and attenuated ventricular dysfunction in response to myocardial ischemia [21]. In response to myocardial injury in mice, cardiac-resident CCR2^+^ macrophages promote monocyte recruitment through a MyD88-dependent production of CCL2, while resident CCR2^−^ macrophages inhibit monocyte recruitment [45]. Further, this study showed that CCR2^+^ resident macrophages are inflammatory, but production of inflammatory cytokines and chemokines is less than recruited CCR2^+^ macrophages. However, resident CCR2^+^ macrophages compared to recruited CCR2^+^macrophages differentially express type I IFN-stimulated genes in response to myocardial injury, which suggests that CCR2^+^-resident macrophages are responsive to type I interferon produced during myocardial infarction.

Circulating monocytes in the mouse consist of two subsets, classical Ly6C^high^ monocytes which are recruited to sites of inflammation and non-classical Ly6C^low^ which patrol the luminal surface of the vascular endothelial cells [138]. Ly6C^high^ and Ly6C^low^ monocyte accumulation in the infracted heart define two distinct sequential phases of monocyte recruitment [8,20] (Table 1). In the initial inflammatory phase 1, Ly6C^high^ monocytes rapidly accumulate in the heart after ischemic injury, reaching a peak at 3 days post infarction and then declining. These monocytes express CCR2 and migrate into injured myocardium in response to the chemokine CCL2 and differentiate into the recruited CCR2^+^ macrophages. These macrophages support inflammation by secreting pro-inflammatory cytokines and chemokines and by secreting matrix metalloproteinases that break down the extracellular matrix in the injured myocardium. These recruited macrophages are phagocytic and remove debris and dying cardiomyocytes, a process that is dependent on the expression of MERKT [139]. Thus, depletion of circulating monocytes with clodronate-loaded liposomes during phase I resulted in larger areas of debris and necrotic tissue, which indicates that influx of Ly6C^high^ monocytes is required for removal of debris and necrotic cells [8].

In the second phase, Ly6C^low^ monocytes/macrophages accumulate in the injured myocardium, with a peak at day 7 post infection (Table 1) and then declining [8,20]. Even though total numbers of Ly6C^low^ macrophages decline after 7 days, 75% of the macrophages are Ly6C^low^ through day 16 [8]. These Ly6C^low^ macrophages are reparative and non-inflammatory. They promote myofibroblast accumulation, angiogenesis and deposition of collagen. Two pathways have been described for the accumulation of Ly6C^low^ macrophages in infracted myocardium. The first pathway involves the migration of circulating Ly6C^low^ monocytes into the myocardium via expression of the chemokine receptor CX_3_CR1, which interacts with the chemokine fractalkine [8]. In this regard, depletion of circulating monocytes during phase II resulted in decreased collagen deposition and reduced numbers of smooth muscle cells. The second pathway is the differentiation of Ly6C^high^ monocytes in the myocardium into proliferating Ly6C^low^ macrophages, which was shown to be dependent on the induction of the orphan nuclear receptor Nr4a1 [20]. In the absence of Nr4a1, Ly6C^high^ monocytes expressed high levels of CCR2, increased mobilization into the myocardium, and differentiated into highly inflammatory macrophages. Further, in the absence of Nr4a1, LV function was impaired and cardiac healing was reduced with increased myocardial scar size and reduced collagen density.

Recent single-cell RNA seq analysis of recruited macrophages in the infarcted myocardium indicate that recruited macrophages are highly plastic and differentiate into multiple subsets with distinct phenotypes [45,46]. Studies by Dick et al. [46] examined recruited macrophages at day 11 after ischemic injury. At this time point the recruited monocytes have differentiated into macrophages. They found three different subsets of resident macrophages populations (described above in Cardiac-Resident Macrophages Section 3 and listed in Table 2) in the steady state and the infarcted heart tissue. In addition, they observed seven macrophage clusters unique to infarcted tissue. These unique clusters represent 64% of the macrophages present in the infracted tissue. Thus, the recruited macrophages differentiate into macrophage subsets with unique transcriptional signatures that are different from the resident macrophage populations. These unique clusters are enriched in both inflammatory pathways and reparative pathways. Single RNA sequencing studies by Bajpai et al. [45] also found seven macrophage clusters after cardiac injury. While the mechanisms involved in the differentiation of recruited monocytes that result in the macrophages clusters are unknown, Bajpai et al. found that depletion of tissue-resident CCR2^+^ macrophages before cardiac injury not only reduced monocyte recruitment and but also decreased a type I Interferon biased cluster. In contrast, depletion of tissue-resident CCR2^−^ macrophages increased numbers of Arg1 and CxCL1 clusters. This suggests that cardiac-resident macrophages are important drivers of the differentiation of recruited monocytes [45]. 

Although recruited macrophages present during the inflammatory phase have been described as M1 macrophages and macrophages present during the reparative phase as M2 macrophages, the transcriptional heterogeneity in recruited macrophages present in infarcted tissues does not easily fit into the MI/M2 paradigm. The M1/M2 nomenclature is derived from studies of in vitro polarization of mouse macrophages with classically activated LPS+IFN-γ (M1) or alternatively activated IL-4 (M2), each with a distinct transcriptome response. See Jablonski et al. [140] for transcriptomes (GSE69607) of classically activated and alternatively activated bone marrow-derived macrophages. Further, research indicated that M1 macrophages have a pro-inflammatory phenotype with anti-pathogen activity while M2 macrophages promote anti-inflammatory and tissue repair responses [141]. Recently Orecchioni et al. [142] found that most surface markers identified on in vitro generated macrophages do not translate to the in vivo situation in classically activated mice. Thus, we prefer that macrophage subsets be classified based on their function and transcriptional profile as suggested by Nahrendorf and Swirski [143]. In this regard, in Table 2 we listed CMs present in the heart during steady state and myocardial infraction based on transcriptional profile and function. 

## 7. Role of Macrophages in Cardiac Remodeling in Hypertension and Diabetic Cardiomyopathy

Diabetes and hypertension, which often co-exist, are predominant risk factors for cardiovascular diseases such as MI and HF, and are characterized by chronic, low grade inflammation, which promotes adverse cardiac remodeling. The inflammatory response is a critical mechanism by which the heart responds to injury and develops adaptive remodeling [144]. However, uncontrolled inflammation can weaken the adaptive response and promote cardiac injury. Both diabetes and hypertension can lead to chronic inflammation and activation of macrophages toward an inflammatory phenotype. Although macrophages play a key role in cardiac remodeling, dysregulation of macrophages between pro-inflammatory and anti-inflammatory phenotypes promotes excessive inflammation and cardiac injury [145]. Cardiac metabolic reprogramming has been implicated in cardiac adaptation to injury [146,147,148], though the healthy heart relies mainly on mitochondrial oxidative phosphorylation metabolism of fatty acid for its energy demands. In contrast, during decompensated HF, the heart relies predominantly on glycolysis [147,149], in which HF leads to the activation of hypoxia-induced factor 1 alpha (HIF-1α) transcription factor which induces transcription of a glycolytic and proinflammatory gene profile [150,151]. HIF1-α can also be stimulated by nonhypoxic mechanisms associated with obesity and hypertension, such as inflammatory cytokines, hyperglycemia, saturated fatty acid activation of TLR4, and oxidized low-density lipoprotein [152,153,154,155]. In addition, diabetic patients are prone to bacterial infections and impaired wound healing, which may be due to the impairments in pro-reparative/anti-inflammatory macrophage functions and increases in a pro-inflammatory macrophage phenotype through enhanced expression of long-chain acyl-CoA synthetase [156,157,158]. Macrophages are also susceptible to protein glycation and formation of advanced glycation end products (ACEs) due to excess glucose levels, which activates the NF-κB pathway and induces production of inflammatory cytokines [159,160]. Additionally, macrophages from patients with coronary artery disease, a common comorbidity of obesity–hypertension, display significantly increased IL-1β and TNF-α expression compared to patients with inflammatory vascular diseases [161]. Therefore, during hyperglycemia, macrophages appear to upregulate glucose uptake and utilization and subsequently increase inflammatory cytokine production, which leads to activation of an inflammatory phenotype [161]. In addition, free fatty acids, lipid mediators and adipokines can promote the inflammatory phenotype [162,163,164].

The source of cardiac macrophages plays a key role in regulation of cardiac remodeling during obesity and hypertension. Generally, the healthy heart contains a low number of M2-like macrophages [45,165,166] which are often considered protective. Recent studies have demonstrated that these M2-like macrophages (embryonic in origin) are lost with aging and replaced with CCR2^+^ monocyte-derived macrophages, even in the absence of injury [45]. In a pressure overload mouse model, resident M2-like macrophages initially promote myocardial adaptive remodeling to mechanical stress. However infiltrating CCR2^+^ monocytes promote maladaptive remodeling during the transition to decompensated HF [167]. The epicardial adipose tissue (EAT) is a rich source of macrophages, neutrophils, and lymphocytes, which may protect the heart from infections [168]. The myokines released from heart muscle during exercise act directly on EAT macrophages to promote an M2-like phenotype and provide protective effects [169]. However, during cardiac injury EAT can be a major source of inflammatory macrophages, due to hypoxia-induced activation of macrophages in EAT which can easily infiltrate into the myocardium [170,171]. In mice, EAT is a major contributor to inflammation during cardiac injury after MI [170], and in MI patients, increased EAT thickness is associated with visceral obesity and cardiac fibrosis [172]. In addition, necrotic myocytes during severe cardiac injury can recruit and activate macrophages via the release of DAMPs [166,173]. The damaged mitochondria in myocytes (a hallmark of HF) can also release DAMPs that enhance the release of proinflammatory chemokines and cytokines [174]. The other major activator of cardiac inflammation during hypertension is endothelial cell injury, due to pressure-induced shear stress and mechanical stretch, which lead to increased reactive oxygen species (ROS) production and impairment of NO signaling, which may attract nearby macrophages [175].

Although the importance of macrophages in cardiac remodeling in myocardial infarction has been studied extensively [46,165,176,177], the role of macrophage metabolism has not been investigated thoroughly. In a mouse model of MI, glycolytic genes, such as GAPDH, are strongly upregulated in macrophages found in the infarcted region at day 1 of MI. In contrast, mitochondrial genes, such as succinate dehydrogenase, are increased at day 3 after MI [173], which suggests that the cardiac microenvironment is hypoxic at day 1 but becomes reoxygenated at day 3 due to vasculogenesis [173,178]. In a rat model of MI, inhibition of glycolysis decreased cardiac macrophage glycolysis and inflammation, and improved LV function [179]. Mechanistically, the efferocytosis of necrotic cells increases intracellular fatty acid supply, fuels mitochondrial fatty acid oxidation and polarizes the macrophage M2-like phenotype [180]. In mice with hypertension and diastolic dysfunction, M2-like polarization contributes to cardiac fibrosis, though this depends initially on inflammatory macrophage infiltration and expansion [181]. Furthermore, in the mouse heart as discussed earlier in this review, resident macrophages, which are negative for CCR2, can be further divided into two groups based on the expression of MHCII [6]. Similarly, CCR2^+^ cardiac monocyte-derived macrophages can be derived from infiltrating Ly6C^hi^ (M1-like) or Ly6C^low^ (M2-like) monocytes [8,167], which suggests that the metabolic profiles may differ not just based on the M1 and M2 paradigm but also on resident macrophages versus infiltrating monocytes and their subsets. Still the immunometabolics field is new and largely based on M1/M2 phenotypes, and in the future research will focus on understanding the metabolic processes of the cardiac macrophage subsets in the remodeling heart.

## 8. Role of Macrophages on Mitochondrial Homeostasis and Collagen Secretion

The majority of the tissues in the body contain tissue-resident macrophages, which are extremely heterogeneous and perform tissue-specific functions—this heterogeneity is dependent on the microenvironment in each tissue. Though macrophages are considered a key cell to mount innate and adaptive immune functions [182], they also mediate tissue-specific functions unrelated to immunity [183]. For example, in the heart, tissue-resident macrophages prevent fibrosis [184], facilitate electrical activity in the atrioventricular node [82] and are involved in healing of injured areas [46,185]. Furthermore, macrophages of unknown function are populated in the myocardium of healthy heart, that yet to be identified. Cardiomycytes are highly specialized cells in the heart with a unique function. Cardiomyocytes are large in size and contain a large number of mitochondria and sarcomeres, which occupy most of the cell volume and deal with the intense metabolic energy and mechanical demands of the heart. Additionally, cardiomyocytes are long lived and rarely renew; they are subjected to the intense mechanical stress and metabolic demands of the beating heart, which requires homeostatic mechanisms to eliminate dysfunctional mitochondria in order to maintain the healthy state of cardiomyocytes. These nonfunctional mitochondria are removed by a process called mitophagy and this preserves the crucial functions of cardiomyocytes [186]. Since cardiomyocytes are subject to extreme energetic demand and the mitochondria produce high energy ATP and reactive oxygen species, it is expected that mitochondria will become damaged and the damaged mitochondria will be degraded via autophagy machinery. Recently, Nicolas-Avila et al. demonstrated a novel non-canonical elimination of dysfunctional mitochondria and other organelles from healthy cardiomyocytes. These authors showed that dysfunctional mitochondria and other cargo are expelled in dedicated membranous particles called exophores. The exophores expose phoshotidylserine on the membrane, which is recognized by the MerTK phagocytic receptor expressed on CMs and the internalized exophores are degraded by phogosome maturation. This intracellular exchange occurs in the heart during homeostasis. In addition, depletion of CMs or deletion of the phagocytic receptor MerTK results in defective elimination of the exophores and mitochondria accumulation in the heart extracellular space, which leads to inflammasome activation, blockage of cardiomyocyte autophagy, accumulation of anomalous mitochondria in cardiomyocytes, metabolic alterations, and ventricular dysfunction [187]. Thus, cardiac macrophages are key for maintaining mitochondrial homeostasis in the healthy heart.

As discussed above, macrophages are highly plastic and functionally diverse during steady state and following injury [188]. Similarly, in the heart tissue, resident macrophages have been implicated in many disease conditions (discussed above). After myocardiac infarction or infection, a large number of immune cells are infiltrated into the heart to remove dying tissue, scavenge pathogens and promote healing. However, in some circumstances, immune cells can cause irreversible damage, contributing to HF. During the initial stage of injury, CMs release pro-inflammatory cytokines to create an inflammatory state, and later transition into a reparative M2-like state, which produce cytokines, chemokines and growth factors (TFG-β and PDGF) that activate fibroblasts to become myofibroblasts, which deposit collagen to form scars. Additionally, CMs produce metalloproteinases and regulate extracellular matrix (ECM) accumulation at the site of injury, which contributes to scar tissue formation [189]. Thus, transition of a tissue repair phase to a regenerative phase following heart injury requires an orchestrated and balanced immune response to fine-tune the interplay between a pro-fibrotic and a pro-regenerative environment. The macrophages are essential to both repair by scar formation and tissue regeneration. However, factors that drive CMs towards either regenerative or pro-fibrotic phenotype are largely unknown [190,191,192,193]. Recently, Simoes et al. [194] demonstrated that the CMs directly contribute to collagen secretion and scar formation in a Zebrafish model and a mouse model. Transcriptomic analysis revealed that the expression of collagen-associated genes (col4a3bpa, col7a1l) was upregulated upon injury in the zebrafish model. In the neonatal mouse model, the authors harvested CMs from P1 and P7 infarcted heart at day 7 and compared gene expression; they reported that several ECM genes Col8a1, Col5a2, Col6a2, Fbn, Postn and Bgn are upregulated, accompanied by evidence of collagen-1+ fibrils in the infarct region. Thus, these findings indicate the CMs are also essential for cardiac tissue repair and regeneration. However, the functional relevance of macrophage-derived collagen production in the context of cardiovascular outcomes remains to be investigated in the future.

## 9. Cardiac-Resident Macrophages and Cardiac Fibrosis

Cardiac fibrosis is an inevitable consequence of chronic insult to the myocardium, characterized by net accumulation of extracellular matrix (ECM) proteins in the cardiac interstitium which leads to wall thickening, systolic and diastolic dysfunction, and impaired overall heart performance. In the healthy heart, the ECM provides a scaffold for cardiac cells and in this manner ensures the structural integrity and function of the heart [189]. In addition, it has been demonstrated that each cardiomyocyte is surrounded by ECM, which ensures the transmission of electrical conduction and contractile force from a single cardiomyocyte into the whole organ, and also ECM function as a reservoir for various latent growth factors [133]. Myocardial injury can be caused by either sterile inflammation or non-sterile inflammation (infection). Moreover, terminating the inflammation requires an anti-inflammatory response, which leads to the initiation of pro-fibrotic signaling [195]. On the other hand, pathophysiological stimuli, for instance pressure overload, volume overload, metabolic dysfunction, and biological aging process may cause interstitial and perivascular fibrosis in the absence of myocardial inflammation.

Both human patients and experimental models of heart disease indicate that the extent of fibrotic remodeling is closely associated with adverse outcome. During the initial inflammatory phase, described above, debris and dead cardiomyocytes are cleared by resident embryonic and recruited macrophages and the extracellular matrix is degraded by secreted matrix metalloproteinases. Once sufficient clearance of necrotic cells and apoptotic immune cells is achieved, a proliferative phase is activated to resolve the damage. During this phase, the fibroblasts trans-differentiate into secretory and contractile cells, called myofibroblasts. This is a key cellular process that results in secretion of an abundant collagen and nonstructural matrix proteins to compensate for the cellular loss in many different conditions associated with HF [133,135]. The proliferative phase is orchestrated by several anti-inflammatory and pro-fibrotic mediators produced by both immune and non-immune cells in the injured myocardium. For example, apoptotic neutrophils release lactoferrin and annexin A1 which inhibit further leukocyte invasion [134], efferocytosis of dead cells stimulates macrophages to produce anti-inflammatory TGF-β1, IL-10, lipoxins and resolvins [134,196,197], regulatory T cell activation leads to additional secretion of TGF-β1 and IL-10 [134], and the secretion of MMPs by immune cells [198] function together and resolve the inflammatory damage. MMPs initiate remodeling by degrading ECM and act to limit leukocyte invasion by cleaving chemokines [199]. TGF-β1 is a well characterized isoform in the cardiac tissue [4], which in the healthy heart is bound to the ECM as an inactive latent complex [200,201]. Cleavage by plasmin and MMP-2 and MMP-9 releases the active form of TGF-β1 [200,202]. Active TGF-β1 binds to its type II receptor (TGFβRII) on the cell surface of fibroblasts and initiates the Smad3- signaling pathways and enhances the expression of a smooth muscle actin (x-SMA) expression and promotes myofibroblast trans-differentiation in the presence of fibronectin domain ED-A [133,203]. These myofibroblasts accumulate in the myocardium in a wide range of pathologic conditions, including myocardial infarction, myocarditis, cardiac pressure or volume overload and alcoholic cardiomyopathy [204,205,206,207]. The activated myofibroblasts secrete elevated levels of collagens and other ECM proteins to maintain the structural integrity and prevent wall rupture and myocardial dysfunction [208]. The excessive deposition of ECM proteins and collagens leads to ventricular stiffness, which can result in contractile dysfunction [209,210], and increase the risk of arrhythmogenesis and mortality [211,212] (Figure 3). Furthermore, myocardial inflammation and fibrosis within the perivascular regions may decrease the oxygen and nutrient availability to cardiac tissue and increase the pathological remodeling response [213]. Following the proliferative stage and establishment of collagen and ECM-based matrix at the infarct site, a maturation stage occurs in which a scar is formed with a cross-linked extracellular matrix. The growth factors and matrix cellular proteins that support the survival and activity of myofibroblasts are depleted, and the majority of the myofibroblasts undergo apoptosis [203,214]. Vascular cells also die, and temporary microvasculature disintegrates. In addition, Lavine et al. demonstrated that tissue-resident macrophages induce cardiomyocyte proliferation and blood vessel development after cardiac injury [42]. Consequently, monocyte recruitment to the injured adult heart is suppressed, and the embryonic macrophage population is largely preserved, resulting in reduced inflammation and accelerated repair. Therefore, cardiac macrophages differentiated from recruited bone marrow-derived monocytes exhibit tissue destructive activity, whereas the embryonically derived tissue-resident macrophage population facilitates the resolution of inflammation and instructs tissue repair in the heart (Figure 3).

## 10. Future Direction of Cardiac Macrophages

Although scientists have identified new phenotypes and functions of cardiac macrophages in the context of MI and HF, their role in cardiac inflammation, tissue regeneration, remodeling and fibrosis is still developing and there remains several important avenues of research for the future. First, although the roles of macrophages in regulating cardiac inflammation, fibrosis, tissue repair and regeneration in different experimental rodent models may overlay, the source and phenotypes of resident macrophage are not completely understood. Recent studies demonstrated that healthy human myocardium is populated with CCR2^−^ macrophages and maintained through local proliferation, while CCR2^+^ macrophages are derived from monocytes and local proliferation [173]. However, the localization and function of these macrophages in the heart is distinct: CCR2^−^ macrophages (M2- like) function as anti-inflammatory, helping to control the inflammation while the CCR2^+^ macrophages (M1-like) initiate the inflammation, which can lead to HF. However, we still lack a clear understanding of whether macrophage function is a cause or consequence in human disease development, and even less is known about the role of these cells after chronic HF has been established. Animal models have provided significant advancement in understanding of the complexity and plasticity underlying resident and recruited cardiac macrophage biology in both steady-state and chronic HF and have identified the importance of macrophages in HF pathogenesis. However, it remains unclear whether these findings will mimic human diseases. Second, the lack of knowledge defining the crosstalk between cardiac-resident macrophages and other cardiac-resident cells such as fibroblasts and cardiomyocytes, and the influence of comorbidities and risk factors on macrophage heterogeneity and function limits our understanding of the role of macrophages in chronic HF, and the development of new therapeutic strategies. Third, macrophage polarization is closely tied to changes in glycolytic (M1-like) and oxidative phosphorylation (M2-like). Macrophage metabolic shifts are not only required for energy demands but also for regulation of pro- and anti-inflammatory processes. Macrophage metabolic shifts are tightly regulated by several intracellular signaling pathways, in which NF-κB, HIF-1α, PDK1 and PPAR-γ play an important role in reprogramming the macrophages [145]. However, investigation pertaining to cardiac macrophage metabolism is still young and requires thorough investigations. Thus, understanding cardiac macrophage metabolism will provide insight into its roles and the mechanisms associated with the remodeling heart and heart failure. Additionally, the thorough investigation of macrophage metabolism will yield unique opportunities to treat HF by targeting macrophage metabolism in the heart without affecting cardiomyocyte metabolism.

## 11. Conclusions

Over the past few years, significant advancements were made to characterize the cardiac immune cells and their role in steady heart function, which will allow for the development of new regenerative targets for HF. In mammalian hearts, the resident cardiac fibroblasts and immune cells, specifically macrophages, regulate cardiomyocyte function in health and disease conditions. The regenerative capacity of cardiomyocytes is very limited in adult hearts and the rate of cardiomyocyte proliferation cannot support the need for new myocytes upon cardiac injury, particularly with infections. However, the resident macrophage populations recognize the danger signal from cardiomyocytes, recruit immune cells and exert detrimental inflammation to clear the infection. At the same time, the immune response activates the cardiac fibroblasts to promote cardiac fibrosis, and cardiac hypertrophy. This leads to further recruitment and enhancement of inflammation, which creates an environment hostile to regeneration and tissue repair. Cardiac fibrosis is beneficial for cardiac tissue repair upon injury. However, uncontrolled tissue damage and continuous activation of pro-inflammatory signaling leads to unregulated extra cellular matrix production, and causes defects in the cardiac electrical conduction system and HF. Despite the incredible research efforts and progress in new drug discoveries that have been made to treat the cardiac dysfunctions during infections or sterile inflammation, the human heart remains unprotected after injury. Therefore, a better understanding of immune cell phenotypes, function, and cardiac macrophage plasticity during cardiac tissue damage and repair could lead to the development of new therapies that improve the outcome of patients with severe myocardial damage caused by either sterile or infectious inflammation.

## Figures and Tables

**Figure 1 cells-10-00051-f001:**
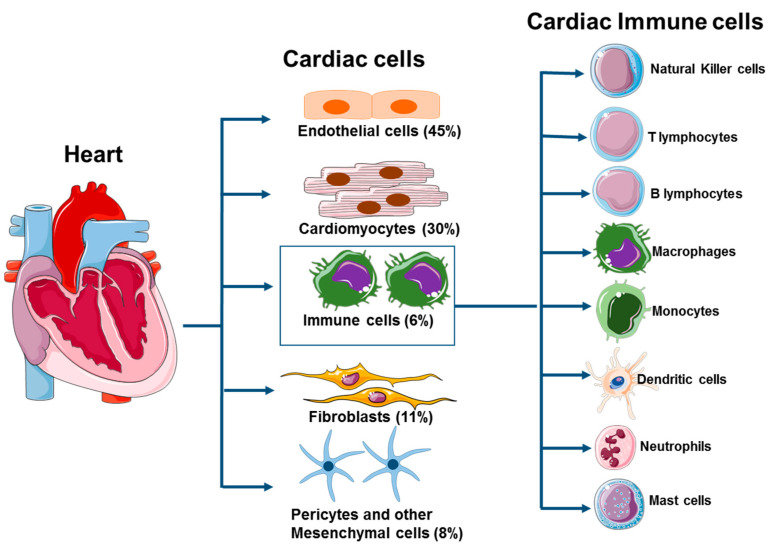
Cellular components of steady-state heart. Heart tissue consists of endothelial cells, myocytes, fibroblasts, pericytes, mesenchymal cells, and various types of immune cells. The macrophages are the major immune cell population in the resting heart and are found in the interstitium and around endothelial cells. Inflammatory monocytes and neutrophils are not present in myocardial tissue but can be observed upon tissue damage. Mast cells, dendritic cells, B cells, NK cells and regulatory T cells are found sporadically in cardiac tissue.

**Figure 2 cells-10-00051-f002:**
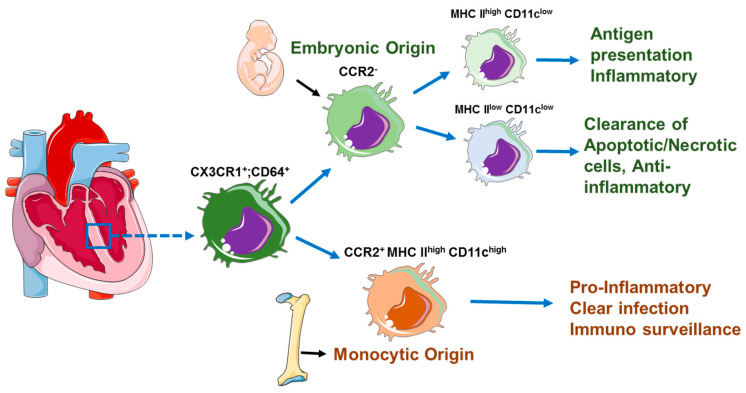
Function of CCR2^+^ and CCR2^−^ cardiac-resident macrophages in the resting heart. An adult heart contains CX3CR1^+^, CD64^+^, CCR2^+^ and CCR2^−^-resident macrophages. The CCR2^−^-resident macrophages enter the heart during embryonic development and possess the capability to self-proliferate. These macrophages are further divided into two groups based on MHC II expression. The CCR2^−^ MHC II^high^ macrophages perform antigen presentation, whereas the CCR2^−^ MHC II^low^ CD11c^low^ macrophage plays an important role in clearance of apoptotic cells and express anti-inflammatory mediators. The CCR2^+^ MHC II^high^ CD11c^high^ originate from bone marrow progenitors and perform immune surveillance functions and protect the heart from infection.

**Figure 3 cells-10-00051-f003:**
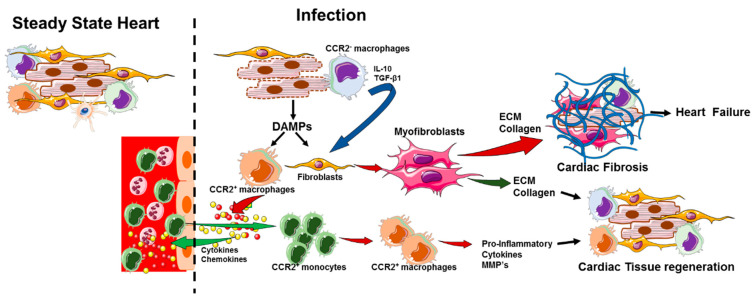
Cardiac immune response during infection and tissue repair. During early infection, cardiomyocytes undergo necrosis and release endogenous danger-associated molecular patterns (DAMPs), which activate resident mast cells and CCR2+ macrophages to release pro-inflammatory cytokines, such as TNFα and IL-1β, and chemokines, which activate endothelial cell dilation and recruit activated CCR2+ monocytes and neutrophils. The CCR2+ monocytes differentiate into macrophages, which together with neutrophils create an inflammatory environment and clear the invaded pathogens. IL-10 and TGFβ1 are released by the action of DAMPs directly on fibroblasts or from CCR2-macrophages that have efferocytosed necrotic cardiomycytes. IL-10 and TGF-β1 then activates fibroblasts to transdifferentiate into myofibroblasts. The myofibroblasts produce extracellular matrix (ECM) and collagen, fill in the gap, and regenerate cardiac tissue. However, increased cardiomyocyte death and cardiac inflammation leads to enhanced myofibroblast activation and uncontrolled production of ECM and collagen which accumulate in the injured heart tissue and cause cardiac fibrosis.

**Table 1 cells-10-00051-t001:** Cardiac immune cells, markers and cell numbers in healthy and ischemic heart.

CD45 + Cells	Markers	Steady State #/mg	Day 1–3 MI #/mg	7 Day MI #/mg	References
Macrophage and DC	CD11b^+^ F4/80^+^ MHC II ^high/low^CD64^+^ Ly6G^−^	567	1000	2000	[7,8]
Total Monocytes	CD11b^+^ F4/80^−^ CD64^−^ Ly6G^−^	80	50,000	40,000	unpublished data, [8]
Classical monocytes	CCR2^high^ Ly6C^high^ CD11c^low^ CX3CR1^low^	25	37,500	10,000	unpublished data, [8]
Non-Classical monocytes	CCR2^high^ Ly6C^low^ CD11c^high^CX3CR1^high^	50	12,500	30,000	unpublished data, [8]
T Cells	CD11b^−^ Ly6G^−^ CD3ε^+^	41	210	110	[7,9]
T Regs	CD4^+^ CD25^+^ Foxp3^+^	60	90	175	[10]
B Cells	CD11b^−^ Ly6G^−^ B220^+^	160	800	100	[9]
Neutrophils	CD11b^+^ Ly6G^+^	18	100,000	5000	[7,8,9]

**Table 2 cells-10-00051-t002:** Cardiac macrophage phenotypes, markers, origin and function.

Cardiac Macrophage Populations	Markers	Origin	Transcriptional Analysis (Cluster Defining Genes)	Function
Leid et al., and Bajpai et al., [43,45]				
C-C chemokine receptor type 2 (CCR2)^−^	CD64+, MERTK+, CX3CR1^high^	Embryonic	Cx3cr1, Lyve, Emr1, CD207, Ccl12, Igf1, Pdgfc, Hbegf, Cyr61	Pro-angiogenic and myogenic, coronary development, and remodeling
CCR2^+^	CD64+, MERTK^+^, CX3CR1^interm^	Monocyte	Ly6C, CXCR2, Sel1, Irf5, Nr4a1	Inflammatory, Type I Interferon response
Epelman et al., [6]				
CCR2^−^ MHCII^high^	CD64^+^, MERTK^+^, CX3CR1^high^, CD206^interm^	Embryonic	MHCII genes, Cd74, Slc11a1, March1, Fcgr2b, Fcgr3, Stab1, Ccl12	Antigen Presentation, Inflammatory
CCR2^−^ MHCII^low^	CD64^+^, MERTK^+^, CX3CR1^interm^, CD206^high^	Embryonic	Fcna, Lyve1, Igf1, Slco2b1, Vsig4, C1qb, Tnk2, Cd164, Snx8, Rab34	Homeostasis, Clearance of necrotic and apoptotic cells
R3 CCR2^+^ MHCII^high^	CD64^+^, MERTK^+^, CX3CR1^high^, CD206^high^	Monocyte	MHCII genes, Cd74, Nlrp3, Nod2, Ptgs2, IL1b, Il18, Mefv, Tnsf14, Rgs1	Antigen Presentation, Pro-inflammatory, IL-1β secretion
Dick et al., [46] Steady-State Populations				
*Timid4* Cluster (Equivalent to CCR2^−^ MHCII^low^)	CCR2^−^, TIMD4^+^, LYVE1^+^, Igf1, MHCII^low^	Embryonic	Timd4, Lyve1, Folr1, Mrc1, Nrp1, Igfbp4, Wwp1, Igf1, Fxyd2, Maf, Gas6	Homeostasis, regenerative functions (endocytosis, lysosome function)
*MHCII* Cluster (Equivalent to CCR2^−^ MHCII^high^)	CCR2^−^, TIMD4^−^, LYVE1^−^, MHCII^high^	Embryonic and Monocyte	MHCII genes, Cd74, Cx3cr1, Axl, Rgs1, Ccl4. Cxcl2, CCl3, Tmsb10, Hexb	Antigen Presentation, Chemokine Response
*Isg* Cluster	CCR2^+^, TIMD4^−^, LYVE1^−^, Irf7, Isg20, Ifit1, MHCII^low^	Monocyte	Irf7, Isg20, Ifit1, C1qa, Stat1, Gbp2, Isg15, Mx1, Xaf1, Ifit2, Ifit3, Oas3, Usp18	Inflammatory, Type I Interferon Response
*CCR2 Cluster* (Equivalent to CCR2^+^ MHCII^high^)	CCR2^+^, TIMD4^−^, LYVE1^−^, MHCII^high^	Monocyte	MHCII genes, Ccr2, Msrb1, Gstp1, Anxa2, Cox5a, Mif, Lgals3, Rac2, Clec12a	Antigen Presentation, Pro-inflammatory, IL-12 and IFN-γ responses
Dick et al. [46] Populations after MI				
HIf1α Cluster	Hif1α	Monocytic	Il1b, S100a11, Cdk2ap2, Msrb1, Adssl1, Tmsb10, Hif1a, Cyp4f18, Plbd1	Response to Hypoxia
cDC2 Cluster	CD209a	Monocytic	Ifitm3, CD209a Samsn1, Tmsb10, Rasgrp2, Nr4a1, Il17ra, Gngt2, Adgre5, Samhd1	Conventional DC cluster
Cluster 6	Ifitm3,	Monocytic	Vegfa, Arg1, Slc7a11, F10, Timp1, Ass1, Cxcl3, Fn1, Clec4e	Extracellular Matrix interactions
Cluster 8	Cd72, IL1b	Monocytic	Cd72, F11r, Sh2d1b1, Napsa, Tmem119, Il1b	Osteoclast Differentiation, Chemokine signaling
Cluster 9	Fcrls, Ccl8	Monocytic	Fcrls, Ccl8	PI-3K signaling, ECM-receptor interactions
Cluster 10	Clec4d, Lrg1	Monocytic	Saa3, Lrg1, Prtn3, Arg1, Fn1, Gda, C4b, Clec4d, Thbs1, Wfdc17, Fam20c	Degradative pathways
Cluster 11	Cd63, Cd9	Monocytic	Gpnmb, Syngr1, Ctsd, Lgals3, Trem2, Ms4a7, Cd63, Fabp5, Clec4d, Cd9 etc.	Lysosome and Glycosaminoglycan degradation, Glutathione metabolism

## Data Availability

Authors do not include any large data sets in this manuscript.

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
