# Peer review of "Role of Cardiac Macrophages on Cardiac Inflammation, Fibrosis and Tissue Repair"

_cells, 2020, doi:10.3390/cells10010051_

Round 1

Reviewer 1 Report

This manuscript is a review of the current literature regarding the role of macrophages in inflammation and remodeling of the heart. Overall, this is a generally well-written and informative review article. I have some suggestions to enhance the manuscript.

Specific Comments:

  1. It would be interesting to have a section on the specific stimuli that activate macrophages in the diseased heart. This is touched on but would be useful to have in more detail. This could be incorporated into the current text if a separate section interrupts the flow of the manuscript.
  2. Line 104, ‘ells’ should be ‘cells’.
  3. All abbreviations should be checked. In the Introduction cardiomyocytes are defined as CM, however, in some later sections CM appears to be used as an abbreviation for cardiac macrophages.
  4. Lines 124 and 125 are separated.
  5. In Figure 2 the MHCIIlow/CD11clow macrophage population is indicated as clearing debri from apoptotic cells, which subtype clears debri from necrotic cells? I think it’s important to indicate this since in ischemia there is widespread necrosis, but also hypertension where there is focal necrosis.
  6. Line 170, ‘tis’ should be ‘this’
  7. Line 257, this paragraph seems out of place.
  8. Line 268, ‘infraction’ should be ‘infarction’. This misspelling occurs in numerous other places in the manuscript.
  9. It would be useful to have more discussion on the contribution of macrophages to adverse remodeling in the hypertensive and diabetic hearts in addition to the discussion already covered for ischemia.
  10. A section on future directions of cardiac macrophage research would also be important.

Author Response

  1. It would be interesting to have a section on the specific stimuli that activate macrophages in the diseased heart. This is touched on but would be useful to have in more detail. This could be incorporated into the current text if a separate section interrupts the flow of the manuscript.

       We would like to thank the reviewer for this valuable suggestion and we               included a section “Activation of cardiac macrophages in diseased heart” to           cover this topic (Line 319 to 364).

  1. It would be useful to have more discussion on the contribution of macrophages to adverse remodeling in the hypertensive and diabetic hearts in addition to the discussion already covered for ischemia.

       We appreciate the reviewers recommendation and in the new version, we             have add a new section “Role of macrophages in cardiac remodeling in                 hypertension and diabetic cardiomyopathy (line 462-532)

  1. A section on future directions of cardiac macrophage research would also be important.

       Thanks for the suggestion and we now added a section for Future direction of         cardiac macrophages (line 658-687).

  4. In Figure 2 the MHCIIlow/CD11clow macrophage population is indicated as            clearing debris from apoptotic cells, which subtype clears debris from necrotic        cells? I think it’s Important to indicate this since in ischemia there is                       widespread necrosis, but also hypertension where there is focal necrosis.

      We agree that necrosis is the major cell death mechanism in cardiac tissue,          we changed the text in the figure 2 and added “Clearance of                                apoptotic/necrotic cells”.

5. All other minor issues are fixed and carefully checked all abbreviations.

Reviewer 2 Report

In their review titled “Role of Cardiac Macrophages on Cardiac  Inflammation, Fibrosis and Tissue Repair’ , the authors discuss the immune cells in heart and focus on role of macrophages in the cardiac injury and repair. While the review is informative, the following changes need to be incorporated

Major revisions

  • In the ‘Cardiac Immune Cells’ section, include a table that lists a) all cardiac immune cells, b) their signature markers (CCR2, CD19/20 etc), c) their population in heart at baseline and d) their population at various significant time points (3 days, 7 days etc.) after ischemic and non-ischemic injury . This table will significantly support this section
  • Include the role of cardiac macrophages in SARS-Cov2 infection and whatever is known till now in the literature
  • Include the role of macrophages in maintaining the mitochondrial homeostasis in heart (https://doi.org/10.1016/j.cell.2020.08.031) as well as their role in collagen secretion
  • Include a separate table that lists the macrophage cell surface markers comparing the M1/M2 classification, and markers used by various labs for identifying different types of macrophages (E.g.: Epelman et al, Dick et al, Nahrendorf et al. Prabhu et al. etc)

Minor revisions

  • The authors used CM interchangeably to refer to cardiomyocytes and cardiac macrophages ( Pg 1, line 41, page 4 – line 114). Fix this
  • In the abstract, the authors define HF as ‘Heart failure (HF) has been conventionally
    14 described as a disease associated with cardiac tissue damage caused by systemic inflammation’. This is an inaccurate statement. HF can be caused by multiple other ‘non-inflammation’ related issues such as arrhythmia and conduction issues.
  • In the cardiac immune cells section, the authors did not include T-cells. Include all the T-cells that play an important role in the heart
  • There are multiple grammatical issues (page 1, line 22 and page 4, line 104 – use of commas vs use of ‘and’), typos (page 4, line 104 – ‘cells, and not ‘ells’, the sentence is missing ‘are’ after macrophages, page 7, like 268 – Myocardial Infarction, not ‘infraction)
  • In figure 1, change the label from ‘Macrophages and Leukocytes’ to ‘Leukocytes’ or ‘Immune cells’. In its current form, this figure is confusing

Author Response

  1. In the ‘Cardiac Immune Cells’ section, include a table that lists a) all cardiac immune cells, b) their signature markers (CCR2, CD19/20 etc), c) their population in heart at baseline and d) their population at various significant time points (3 days, 7 days etc.) after ischemic and non-ischemic injury . This table will significantly support this section

      It is a great suggestion, we created a table and included all required                      information. Thanks to the reviewer.

  1. Include the role of cardiac macrophages in SARS-Cov2 infection and whatever is known till now in the literature

       As per reviewer’s’ suggestion, now we added a paragraph in section 4 (lines           217-251).

  1. Include the role of macrophages in maintaining the mitochondrial homeostasis in heart (https://doi.org/10.1016/j.cell.2020.08.031) as well as their role in collagen secretion.

        We thank reviewer for bringing this important point, in the current version            we included a section “Role of macrophages on mitochondrial homeostasis            and collagen secretion” (line 534-587)

  1. Include a separate table that lists the macrophage cell surface markers comparing the M1/M2 classification, and markers used by various labs for identifying different types of macrophages (E.g.: Epelman et al, Dick et al, Nahrendorf et al. Prabhu et al. etc)

       We appreciate the reviewer for this valuable suggestion, and now we have             made a table and included in the text (Table 2). Since, Prabhu et al.’s review         article talks about various cell types, we discussed Dr. Prabhu’s findings in             the review.

Minor revisions

  1. The authors used CM interchangeably to refer to cardiomyocytes and cardiac macrophages (pg. 1, line 41, page 4 – line 114). Fix this

       We are sorry for the mistake. We have taken serious actions and carefully             reviewed the text and corrected the abbreviation (Cardiac macrophages as           CM and we spelled out cardiomyocytes)

  1. In the abstract, the authors define HF as ‘Heart failure (HF) has been conventionally14 described as a disease associated with cardiac tissue damage caused by systemic inflammation’. This is an inaccurate statement. HF can be caused by multiple other ‘non-inflammation’ related issues such as arrhythmia and conduction issues.

       We agree with the reviewer and we changed the text in the abstract to                 accommodate arrhythmia and conduction issues associated with HF.

  1. In the cardiac immune cells section, the authors did not include T-cells. Include all the T-cells that play an important role in the heart

        We appreciate the comment and we added the text to the cardiac immune            cells section (Line 101-106).

  1. There are multiple grammatical issues (page 1, line 22 and page 4, line 104 – use of commas vs use of ‘and’), typos (page 4, line 104 – ‘cells, and not ‘ells’, the sentence is missing ‘are’ after macrophages, page 7, like 268 – Myocardial Infarction, not ‘infraction)

       We are sorry for the grammatical errors in the manuscript, we have carefully         read the manuscript and fixed all grammatical errors. Thanks.

  1. In figure 1, change the label from ‘Macrophages and Leukocytes’ to ‘Leukocytes’ or ‘Immune cells’. In its current form, this figure is confusing

    Thanks for the suggestion and we changed the ‘Macrophages and Leukocytes’ in to Immune cells.

Round 2

Reviewer 2 Report

In table 1, the authors mention 'Classical' and 'Non-classical'. Are these macrophages? They need to mention the cell type here

In table 1, the authors label Neutrophils as CD11b negative and refer to Nahrendorf et al (reference 16). This is inaccurate. in the 'Materials and Methods', 'mAbs and flow cytometry' section, they define Neutrophils as CD11bhi (CD90/B220/CD49b/NK1.1/Ly-6G)hi (F4/80/I-Ab/CD11c)lo Ly-6Cint

One of the challenges in this field is the discrepancy/diversity amongst the macrophage markers used by various research groups, to label cardiac macrophages. In this review, the authors have an opportunity to discuss most commonly used cardiac macrophage markers reported in literature and highlight the need to standardize this aspect. In table 2, please also include the macrophage markers reported by groups like Douglas Mann, S.Prabhu, Frangogiannis as well, and highlight the similarities and differences in markers used by the major groups. The inclusion of the recent sc-RNA seq data by Dick et al. is appreciated.

Also, please explain what R1, R2 and R3 are, in table 2

Fix the typos - Epleman to Epelman (Table 2), Herat Failure to Heart Failure (Figure 3)

Author Response

  1. In table 1, the authors mention 'Classical' and 'Non-classical'. Are these macrophages? They need to mention the cell type here

Thanks for the suggestion and these cells are monocytes not macrophages, and the current version we clearly marked it. 

  1. In table 1, the authors label Neutrophils as CD11b negative and refer to Nahrendorf et al (reference 16). This is inaccurate. in the 'Materials and Methods', 'mAbs and flow cytometry' section, they define Neutrophils as CD11bhi (CD90/B220/CD49b/NK1.1/Ly-6G)hi (F4/80/I-Ab/CD11c)lo Ly-6Cint.

Sorry for the wrong information, in the current version we changed in to CD11b positive. Thanks

  1. One of the challenges in this field is the discrepancy/diversity amongst the macrophage markers used by various research groups, to label cardiac macrophages. In this review, the authors have an opportunity to discuss most commonly used cardiac macrophage markers reported in literature and highlight the need to standardize this aspect. 

Thanks a lot for this important point, and in our current version we took the opportunity to explain the importance of this review.

  1. In table 2, please also include the macrophage markers reported by groups like Douglas Mann, S.Prabhu, Frangogiannis as well, and highlight the similarities and differences in markers used by the major groups. The inclusion of the recent sc-RNA seq. data by Dick et al. is appreciated.

We appreciate the reviewer’s comment about the macrophage markers and suggestions to include the experts (D. Mann, S. Prabhu and Frangogiannis) work in this field. We would like to inform the reviewer that we have discussed the published work from S. Prabhu and Frangogiannis et al (review article) in the text. The S. Epelman paper that we are referring a lot is from Douglas Mann’s group. The other paper from this same group (Lavine et.al.) described the macrophage population in neonatal and adult hearts. We have discussed this CM phenotypes and markers in the text (lines 378 to 389) and included the adult CM phenotype in table 2. We have also included recent work from the Lavine lab (Bajpai et al.) in table 2 and discussed the single cell RNA seq data from this paper in the text (lines 459-456)     

Thus, we discussed all the current literature related to the CMs in this review. 

  1. Also, please explain what R1, R2 and R3 are, in table 2

The R1, R2 and R3 populations are based on the marker expression profile, now in current version we explained more clearly by removing the R1-R3 designations and referring to the marker expression profile, eg. CCR2- MHChigh.

Fix the typos - Epleman to Epelman (Table 2), Herat Failure to Heart Failure (Figure 3)

We apologies for the mistakes, in this current version we fixed these errors.